# Adding Anti-HER2 Therapy to Neoadjuvant Endocrine Therapy Seems Effective in Hormone Receptor and HER2 Positive Breast Cancer Patients Unfit for Chemotherapy: A Nationwide Population-Based Cohort Study

**DOI:** 10.3390/cancers16244188

**Published:** 2024-12-16

**Authors:** Anne de Bruijn, Robert-Jan Schipper, Adri C. Voogd, Marleen J. J. Pullens, Johanne G. Bloemen, Linda de Munck, Yvonne E. van Riet, Sabine Siesling, Birgit E. P. Vriens, Grard A. P. Nieuwenhuijzen

**Affiliations:** 1Department of Surgery, Catharina Hospital Eindhoven, 5623 EJ Eindhoven, The Netherlands; info@rjschipper.nl (R.-J.S.);; 2Department of Research and Development, Netherlands Comprehensive Cancer Organization (IKNL), 3511 LC Utrecht, The Netherlands; 3Maastricht University Medical Center, GROW—School for Oncology and Developmental Biology, 6229 HX Maastricht, The Netherlands; 4Master Advanced Nursing Practice, Faculty of People and Health, Fontys University of Applied Sciences, 5022 DM Tilburg, The Netherlands; 5Department of Health Technology and Services Research, Technical Medical Centre, University of Twente, 7522 NB Enschede, The Netherlands; 6Department of Internal Medicine, Catharina Hospital Eindhoven, 5623 EJ Eindhoven, The Netherlands

**Keywords:** neoadjuvant endocrine therapy, HER2-positive breast cancer, neo-adjuvant targeted therapy, elderly patients

## Abstract

Data are lacking on the optimal neoadjuvant systemic treatment of patients with hormone receptor and HER2 positive breast cancer who are unfit for the combination of chemotherapy and anti-HER2 therapy. A nationwide series of 190 breast cancer patients treated with neoadjuvant endocrine treatment (NET) or NET+anti-HER2 were evaluated. The ypT0 rate was significantly higher after NET+anti-HER2, with 10.0% (4/40) versus 1.3% (2/150) following NET (*p* = 0.019). The ypN0 rate was significantly higher after NET+anti-HER2, with 25.0% (6/24) versus 5.5% (3/55) following NET in cN+ patients (*p* = 0.020) and 81.3% (13/16) versus 55.8% (53/95) after NET in cN- patients (*p* = 0.047). In cN- patients, ypN0 status was independently associated with age (*p* = 0.008) and administration of NET+anti-HER2 (*p* = 0.016). These findings suggest a clinical benefit of treatment with NET+anti-HER2 if patients are unfit for the combination of anti-HER2 therapy with chemotherapy.

## 1. Introduction

In the Netherlands, about 15,000 patients are treated with curative intent for early-stage invasive breast cancer every year [1]. Based on molecular subtypes, there are the following four different categories of breast cancer: Luminal A, Luminal B, human epidermal growth factor receptor 2 positive (HER2+), and the basal-like type. Only fifteen percent of newly diagnosed breast cancers are HER2+. HER2+ tumours are more aggressive and associated with a worse overall survival than HER2-negative breast cancer [2]. However, survival has dramatically improved with the discovery of anti-HER2 therapy (i.e., trastuzumab) when administered in addition to standard chemotherapy [3,4].

Initially, anthracyclines were part of the standard chemotherapy regimens. Such regimens were known for their toxicity, both in the short and the long term, especially in elderly patients [5]. More recently, regimens without anthracyclines were also proven to be effective in patients with small, node-negative HER2+ breast cancer [6,7]. To minimize short- and long-term morbidity even further, a possible strategy would be to omit chemotherapy and provide anti-HER2 therapy alone or in combination with endocrine therapy (ET) in hormone receptor-positive (HR+) breast cancer, both in the neoadjuvant and adjuvant setting. Reducing morbidity is of special interest in patients who are unfit for chemotherapy, mostly due to higher age or comorbidities. In addition, the incidence of HR+/HER2+ breast cancer seems to be rising in the elderly population [8]. As a result, clinicians are more often confronted with this treatment dilemma. Currently, elderly patients with HR+/HER2+ breast cancer are most frequently treated with ET alone, without anti-HER2 therapy. The question is whether the added toxicity of anti-HER2 is justified in the neoadjuvant setting for patients with HR+/HER2+ breast cancer who are unfit for chemotherapy.

In different settings, the previous studies aimed to improve the overall survival rate by combining ET with anti-HER2 therapy. For example, in the metastatic setting, the TAnDEM and PERTAIN trials suggested that the combination of an aromatase inhibitor and anti-HER2 therapy led to significantly improved overall survival compared to anti-HER2 therapy alone [9,10].

In the adjuvant setting, a retrospective study from the United States compared ET versus ET in combination with anti-HER2 therapy and found no overall survival benefit for the combination of ET with anti-HER2 therapy in HR+/HER2+ patients [11].

In the neoadjuvant setting, many studies have reported on the effectiveness of only anti-HER2 therapy. In the NeoSphere study, patients randomized to dual anti-HER2 therapy (trastuzumab + pertuzumab) without chemotherapy achieved a pathological complete response (pCR) rate of 16.8%. This rate was significantly lower than the pCR rate of 45.8% that was achieved by the group in which anti-HER2 therapy was combined with chemotherapy [12]. Two other prospective studies (TBCRC 006, PAMELA trial) reported on the combination of lapatinib + trastuzumab as a form of anti-HER2 monotherapy, which resulted in pCR rates in the breast of 23 to 26% [13,14].

In all these neoadjuvant trials, the patients were treated with anti-HER2 therapy without the use of synchronous ET. Hence, it is unknown whether the use of ET combined with anti-HER2 therapy (ET+aHER2) is effective in the neoadjuvant setting. Consequently, data are lacking on the optimal neoadjuvant systemic treatment of patients with HR+/HER2+ breast cancer who are unfit for the combination of chemotherapy and anti-HER2 therapy.

The primary aim of this study is to determine whether the ypT0 and ypN0 rates of the breast and axilla differ between patients treated with neoadjuvant endocrine treatment (NET) combined with anti-HER2 therapy (NET+aHER2) than in patients treated only with NET. The secondary aim is to determine whether there is an association between patient characteristics, clinical tumour characteristics, and treatment of choice.

## 2. Materials and Methods

The present study is a quantitative retrospective nationwide population-based cohort study. Data were obtained from the Netherlands Cancer Registry (NCR), which is hosted by the Netherlands Comprehensive Cancer Organization (IKNL). The NCR comprises data of all the patients diagnosed with breast cancer in the Netherlands. Since all the data stored in the NCR are anonymized, the need for informed consent was waived. For this study, patients diagnosed with primary invasive breast HR+/HER2+ cancer between 2008 and 2019 and treated with NET or NET+aHER2 were selected. Patients were included if they had been treated for at least 28 days with NET. Patients were excluded if they had received neoadjuvant chemotherapy or neoadjuvant chemotherapy in combination with anti-HER2 therapy, if they were male, and if they were younger than 18. After neoadjuvant systemic treatment, patients could be surgically treated with a lumpectomy or mastectomy, followed by either no axillary surgery, sentinel lymph node biopsy, targeted axillary dissection, or axillary lymph node dissection.

### 2.1. Treatment

During the study period, different updates of the Dutch national guidelines were applicable [15,16,17]. In short, for patients with HER2+ breast cancer and with an indication for systemic therapy, the guideline suggested administering systemic therapy (chemotherapy in combination with anti-HER2 therapy) in the neoadjuvant setting. However, the guidelines gave no advice for HER2+ patients who were unfit for chemotherapy. Theoretically, there would be the following two treatment options in such cases: NET alone or NET+aHER2. Regarding endocrine treatment, the guideline indicated that premenopausal patients were to be treated with tamoxifen 20 mg once daily in combination with a gonadotropin-releasing hormone (GnRH) analogue. Postmenopausal patients were to be treated either with tamoxifen 20 mg once daily or with an aromatase inhibitor of choice. The guideline did not provide any recommendations for the duration of NET, nor for the frequency and techniques of choice for treatment evaluation during NET. In patients treated with anti-HER2 therapy, trastuzumab was administered once every 3 weeks.

### 2.2. Pathology

The core biopsies and surgical specimens were evaluated according to the applicable Dutch breast cancer guidelines [15,16,17]. First, a core biopsy of the primary tumour at diagnosis was evaluated using standard haematoxylin and eosin staining. In addition, immunohistochemistry or fluorescence (chromogenic) in situ hybridization was performed to obtain the histological subtyping, oestrogen and progesterone receptor status (positive if ≥10%), and HER2 status. During the study period, TNM 6 was used until 2009, TNM 7 was used between 2010 and 2016, and TNM 8 was used after 2016. The ypT0 and ypN0 statuses were determined if no invasive or in situ tumour cells were left after the neoadjuvant systemic therapy.

### 2.3. Statistical Analyses

Descriptive analyses were used to summarize patient and treatment characteristics. Normally distributed data were reported as mean with standard deviation. Non-normally distributed data were reported as median with range. Patients were divided into the following two groups: (1) treated with NET alone, or (2) treated with NET+aHER2. Medians were compared between these two groups using the Mann–Whitney U test. Additionally, trends between the patient characteristics, clinical tumour characteristics, and treatment of choice were analysed with crosstabs using the chi-squared test or with Fisher’s exact test, if applicable.

Furthermore, the associations were analysed among the patient, the tumour and treatment characteristics, and the achievement of ypT0 in the breast or ypN0 in the axilla (all as categorical dichotomous variables) using uni- and multivariable logistic regression analyses. The covariates that were independently associated with ypT0 and ypN0 were identified using stepwise regression with backwards selection. The multivariable regression equation only included covariates with *p* < 0.25 in the univariable analyses. Data were analysed using SPSS version 28 (SPSS Inc., Chicago, IL, USA). A *p*-value of *p* < 0.05 was considered statistically significant.

## 3. Results

### 3.1. Patients

Between 2008 and 2019, 15,736 patients were diagnosed with HR+/HER2+ breast cancer. A total of 190 patients were included (see Figure 1). Patient, tumour, and treatment characteristics are summarized in Table 1. Median age at diagnosis was 77 years (range 37–98). Almost all patients (182/190, 95.8%) were post-menopausal, and most had invasive breast cancer of no special type. In 39.5% of the patients, ER+ was combined with PR-. The performance status of the patients was available for 58 patients, 9 of whom were scored as WHO-2, 1 as WHO-3, and 1 as WHO-4.

### 3.2. Treatment

In total, 150 (78.9%) patients were treated with NET only, and 40 (21.1%) were treated with NET+aHER2. NET consisting of an aromatase inhibitor was administered to 132 (69.5%) patients, whereas 25 (13.2%) received tamoxifen (+/− GnRH). No details on the type of prescribed NET were available in 33 (17.3%) patients. In all 40 patients treated with NET+aHER2, anti-HER2 therapy consisted of trastuzumab monotherapy. The median duration of neoadjuvant systemic treatment was 176 days (range 29–1040). In the NET group, the median duration was significantly shorter, with 156 days compared to 189 days in the NET+aHER2 (*p* = 0.002). A mastectomy was performed in 116 (61.1%) patients and a lumpectomy in 74 (38.9%) patients. No surgical staging of axillary lymph nodes was performed in 18 of the 190 patients (9.5%). Significantly more patients with clinically node-positive (cN+) disease were treated with NET+aHER2 (*p* = 0.029) (see Table 1).

### 3.3. ypT and ypN Status

Overall, ypT0 status was observed in six (3.2%) patients, two of whom were treated with NET and four of whom were treated with NET+aHER2. The ypT0 rate was significantly higher in patients treated with NET+aHER2 than in patients treated with NET (*p* = 0.019); of the patients treated with NET, 1.3% (2/150) experienced ypT0 versus 10% (4/40) of the patients treated with NET+aHER2 (see Table 2).

Overall, ypN0 status was observed in 75 (39.5%) patients, 56 of whom were treated with NET and 19 of whom were treated with NET+aHER2. The ypN0 rate did not significantly differ between the treatment groups (*p* = 0.162); of the patients treated with NET, 37.3% (56/150) experienced ypN0 versus 47.5% (19/40) of the patients treated with NET+aHER2.

Of the 111 patients with cN−, 95 (85.6%) were treated with NET and 16 (14.4%) with NET+aHER2. The ypN0 rate was significantly higher in patients treated with NET+aHER2, with 81.3% (13/16) versus 55.8% (53/95) of the patients treated with NET (*p* = 0.055) (see Table 3).

Of the 79 patients with cN+, 55 (69.6%) were treated with NET and 24 (30.4%) with NET+aHER2. The ypN0 rate was significantly higher in patients treated with NET+aHER2, with 25% (6/24) versus 5.5% (3/55) of the patients treated with NET (*p* = 0.020) (see Table 4). Further details about ypT status in relation to ypN status are presented in Table 5.

### 3.4. Factors Associated with ypN0 Status

In cN- patients (N = 111), univariable analyses suggested the ypN0 status to be associated with age, grade, duration neoadjuvant treatment, and type of neo-adjuvant treatment (*p* < 0.25). In the multivariable analysis, age < 75 years and NET+aHER2 were associated with a significantly higher probability of having ypN0 status. Further details are summarized in Table 6.

In cN+ patients (N = 79), univariable analyses showed that the ypN0 status tended to be associated with the type of neoadjuvant treatment (*p* = 0.020). No multivariable analysis was performed for this group, since ypN0 status was observed in only nine patients.

## 4. Discussion

The aim of this nationwide population-based retrospective cohort study was to determine whether the rates of ypT0 and ypN0 are higher in patients treated with NET+aHER2 than in patients treated with NET. Of the 190 patients included with HR+/HER2+ breast cancer, 150 were treated with NET and 40 with NET+aHER2. Patients with cN+ breast cancer were significantly more frequently treated with NET+aHER2 than with NET. Patients treated with NET+aHER2 were significantly more likely to experience ypT0 status. Also, ypN0 status was observed significantly more often among patients treated with NET+aHER2 than among patients treated with NET, both in those with cN− and cN+ statuses.

Only 1.3% of the patients treated with NET in the present study experienced a pCR in the breast (ypT0) compared to 10.0% in the patient group treated with NET+aHER2. This finding is in line with the results of the randomized controlled NeoSphere study, in which 51 patients with HR+/HER2+ breast cancer were treated with trastuzumab and pertuzumab without chemotherapy or ET, resulting in a ypT0 rate in the breast of 5.9% [12]. It is difficult to determine whether this ypT0 rate of 5.9% is significantly lower than the 10.0% found in the present study, since patients in the NeoSphere study were younger and their tumour characteristics were less favourable than in the present study. Nonetheless, the present study confirms that the probability of having ypT0 status of the breast is low in HR+/HER2+ breast cancer patients treated with neoadjuvant systemic treatment without chemotherapy.

The question arises of whether the administration of anti-HER2 therapy as an addition to ET in the curative setting is effective regarding overall survival, since a retrospective analysis from the United States based on data derived from the National Cancer Database (NCDB) showed no significant difference in survival with the addition of anti-HER2 therapy to ET in the adjuvant setting in patients who did not receive chemotherapy [11]. Nonetheless, studies in the metastatic setting showed that progression-free survival was prolonged by the combination of an aromatase inhibitor with anti-HER2 or dual anti-HER2 [10]. It is likely that no improvement in overall survival following the addition of anti-HER2 in the adjuvant setting could be reported since the patients included were at low risk for tumour recurrence [11]; more than 77% of the patients included had a pT1 tumour, and almost 30% had a pT1a tumour. In such tumours, the effectiveness of administering any kind of systemic therapy is questionable, though not negligible [18,19].

A major strength of the present study is that all patients had an indication for adjuvant systemic therapy based on their tumour stage (all were >pT1a or N+). Therefore, it is plausible that overall survival could have improved in patients with ypT0 or ypN0. Furthermore, all these observations underline the need for adequate patient selection before drawing conclusions regarding the effectiveness of anti-HER2 therapy in addition to ET.

In patients with cN+ disease, the ypN0 rate was significantly higher in patients treated with NET+aHER2 than in those treated with NET. Formerly, patients with cN+ breast cancer were treated with an axillary lymph node dissection, which is associated with significant morbidity, such as lymphoedema [20,21]. Nowadays, patients with ypN0 are identified with several minimally invasive surgical restaging protocols, such as marking the axilla with a radioactive iodine seed (MARI) or targeted axillary dissection (TAD) [22,23,24], both as standalone procedure or in combination with sentinel lymph node biopsy. In these patients, an axillary lymph node dissection is avoided, which will reduce the associated morbidity [25]. Although the follow-up data are still immature, available evidence has reported excellent low axillary recurrence rates [25,26,27,28,29,30]. The addition of anti-HER2 therapy to NET can minimise the extent of axillary surgery thus improving the quality of life of these elderly patients.

The present study reported a possible positive effect of the addition of anti-HER2 therapy to NET. Although future studies should confirm its effect on overall survival, concerns may arise of whether patients who are unfit for chemotherapy can tolerate anti-HER2 therapy. The prospective, randomized, controlled RESPECT trial compared trastuzumab monotherapy with chemotherapy in combination with trastuzumab in the adjuvant setting in patients older than 70 years [31]. In the trastuzumab monotherapy group, the trial reported no grade 4 hematologic adverse events. Nonetheless, grade 3 neutropenia occurred in 9.6% of patients, while non-haematological grade 3 toxicity was reported in 11.9%. Finally, none of the patients discontinued trastuzumab monotherapy [31]. However, a clinically significant decline in the left ventricle ejection fraction was observed in 8.1%. These results were in line with the findings of the NeoSphere study. Moreover, the quality of life of the patients treated without chemotherapy was significantly better. These findings all indicate the relatively mild toxicity of anti-HER2 therapy in the elderly breast cancer patient, suggesting that the combination of NET+aHER2 is well tolerated, similarly for most of the patients included in the present study.

The following limitations apply to the present study. Firstly, although nationwide, the study population is still small, with a total of 190 patients, only 40 of whom received NET+aHER2. This could have led to clinically relevant differences remaining undetected (i.e., type II errors). Consequently, the number of patients with ypT0 in the breast or a ypN0 in the axilla was small, which limited the possibilities of performing adequate multivariable analyses. Furthermore, data on WHO performance status were missing for 70% of the patients, and no data were available on cardiac function. As a result, it was not possible to objectively determine whether patients who did not receive aHER2 were fit for treatment with anti-HER2 agents. This probably led to confounding by indication. Finally, due to missing breast cancer-specific follow-up data, in-depth survival analyses were not performed. This limitation is of special interest since the use of pCR as a validated surrogate endpoint for overall survival is under debate, especially in patients with HR+/HER2+ breast cancer [32,33,34,35]. In addition, it cannot be ruled out that bias occurred, given the long period of inclusion in this retrospective analysis (2008–2019), as there may have been changes in the applicable guidelines during study period regarding diagnostic workup and treatment.

With all these limitations in mind, the current study is intended more to generate hypotheses for prospective studies rather than changing practices. More research seems justified to identify patients unfit for chemotherapy who can benefit from the addition of anti-HER2 therapy to ET. In the current study, “unfit” is somehow a surrogate for the elderly patient, since the median age of 77 years was reported. These future studies should include a larger study population than the present study. Secondly, patients should preferably be treated with a dual-HER2 blockade, which seems beneficial even in combination with endocrine therapy [36]. The combination of anti-HER2 therapy with endocrine therapy may overcome both endocrine and anti-HER2 resistance [37]. This hypothesis is supported by the results of the interim analysis of the WSG-ADAPT phase II trial, which demonstrated pCR rates of more than 40% after short dual-HER2 blockade using trastuzumab emtansine combined with endocrine therapy, without systemic chemotherapy, in HR+/HER2+ early-stage breast cancer [36]. Furthermore, the quality of life of patients treated with NET+aHER2 should be monitored, since anti-HER2 therapy in elderly patients also has side effects. However, these side effects are less common than the side effects found in patients treated with chemotherapy [31]. Lastly, the RESPECT trial demonstrated that patients older than 70 years tolerated anti-HER2 therapy without severe complications [31]. Nonetheless, it is still necessary to monitor the cardiac performance status [38].

## 5. Conclusions

In conclusion, the ypT0 and ypN0 rates in patients with HR+/HER2+ breast cancer treated with neoadjuvant therapy were significantly higher when treated with anti-HER2 therapy in addition to NET. This finding suggests a clinical benefit of treatment with NET+aHER2 if patients are unfit for the combination of anti-HER2 therapy with chemotherapy, especially in cN+ patients. Further research is warranted to determine if these higher ypT0 and ypN0 rates translate into improved breast cancer-specific and overall survival in elderly HR+/HER+ breast cancer patients.

## Figures and Tables

**Figure 1 cancers-16-04188-f001:**
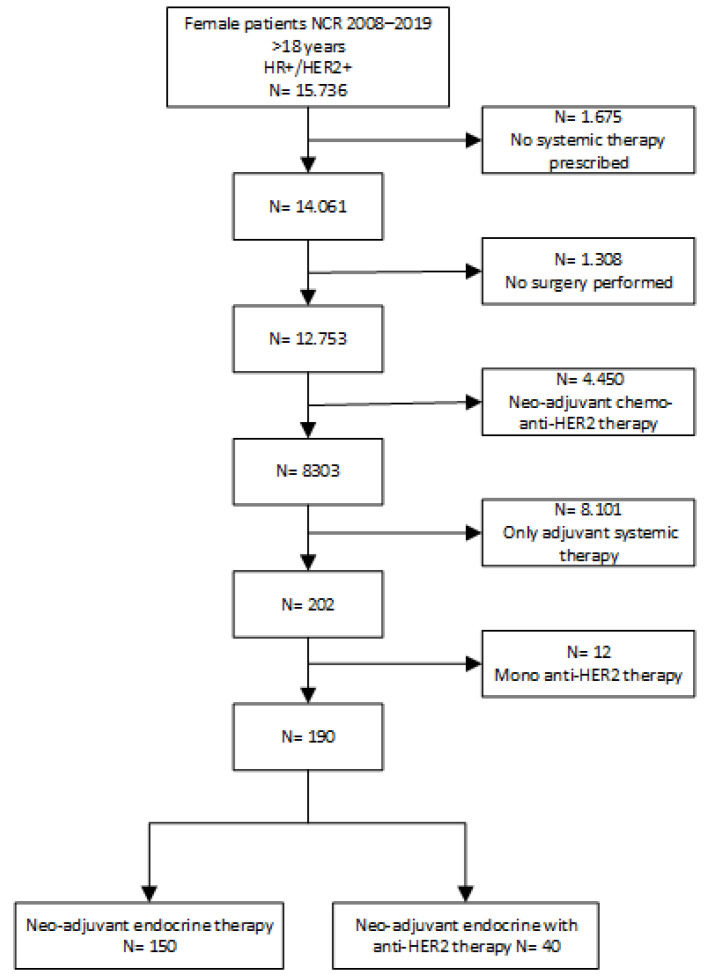
Selection of patients.

**Table 1 cancers-16-04188-t001:** Patient and neoadjuvant treatment characteristics.

		Total	NET	%	NET+aHER2	%	*p*-Value
Patients (N)		190	150	79.0	40	21.0	
Age years							0.734 ^1^
	<50	6	4	66.6	2	33.4	
	50–75	60	47	78.3	13	21.7	
	>75	124	99	79.8	25	20.2	
Menopausal status							0.368 ^2^
	Pre/peri	8	5	62.5	3	37.5	
	Post	182	145	79.7	37	20.3	
Histologic subtype							0.357 ^1^
	NST	164	130	79.3	34	20.7	
	Lobular	21	15	71.4	6	28.6	
	Other	5	5	100.0	-	0.0	
Clinical tumour status							0.093 ^1^
	cT1	40	35	87.5	5	12.5	
	cT2	94	77	81.2	17	18.8	
	cT3	16	11	68.8	5	31.2	
	cT4	40	27	67.5	13	23.5	
Clinical nodal status							0.029 ^1^
	cN0	111	95	85.5	16	14.5	
	cN1	72	50	69.4	22	30.6	
	cN2/3	7	5	71.4	2	28.6	
Multifocality							0.176 ^1^
	Unifocal	148	120	81.0	28	19.0	
	Multifocal	42	30	71.4	12	28.6	
Receptor status							0.421 ^1^
	ER + PR + HER2+	115	93	80.8	22	19.2	
	ER + PR − HER2+	75	57	76.0	18	24.0	
Grade							0.977 ^1^
	1	12	10	83.3	2	16.7	
	2	70	55	78.5	15	21.5	
	3	40	31	77.5	9	22.5	
	Missing	68	54	79.4	14	20.6	
DCIS component							0.460 ^1^
	Not present	128	103	80.4	25	19.6	
	Present	62	47	75.8	15	24.2	
Type NET							0.632 ^1^
	Tamoxifen (+/−GnRH)	25	18	72.0	7	28.0	
	Aromatase inhibitor	132	105	79.5	27	20.5	
	Missing	33	27	81.8	6	18.2	
Duration NET days							0.027 ^1^
	<180	97	84	86.6	13	13.4	
	181–270	66	46	70.0	20	30.0	
	>270	27	20	74.1	7	25.9	
Targeted Therapy							n.a.
	Trastuzumab	40	-	0.0	40	100.0	

NST = No special type, ER = oestrogen receptor, PR = progesterone receptor, HER2 = human epidermal growth factor receptor 2, DCIS = ductal carcinoma in situ, NET = neoadjuvant endocrine therapy, NET+aHER2 = neoadjuvant endocrine + anti-HER2 therapy, GnRH = gonadotropin-releasing hormone, ^1^ = chi-squared test, ^2^ = Fisher’s exact test. n.a. = not applicable.

**Table 2 cancers-16-04188-t002:** Comparison of factors of patients with and without ypT0 status.

		Total	%	ypT0	%	Non ypT0	%	*p*-Value
Patients (N)		190	100	6	3.2	184	96.8	
Age years								0.903 ^1^
	<50	6	3.3	0	0	6	3.3	
	50–75	60	31.5	2	33.3	58	31.5	
	>75	124	65.2	4	66.7	120	65.2	
Menopausal status								1.000 ^2^
	Pre/peri	8	4.3	0	0	8	4.4	
	Post	182	95.7	6	100	176	95.7	
Histologic subtype								0.841 ^1^
	NST	164	86.4	5	83.3	159	86.4	
	Lobular	21	10.9	1	16.7	20	10.9	
	Other	5	2.7	0	0	5	2.7	
Clinical tumour status								0.796 ^1^
	cT1	40	21.2	1	16.7	39	21.2	
	cT2	94	49.5	3	50	91	49.5	
	cT3	16	8.7	0	0	16	8.7	
	cT4	40	20.7	2	33.3	38	20.7	
Clinical nodal status								0.762 ^1^
	cN0	111	58.7	3	50	108	58.7	
	cN1	72	37.5	3	50	69	37.5	
	cN2/3	7	3.8	0	0	7	3.8	
Multifocality								1.000 ^2^
	Unifocal	148	77.7	5	83.3	143	77.7	
	Multifocal	42	22.3	1	16.7	41	22.3	
Receptor status								0.003 ^2^
	ER + PR + HER2+	115	62.5	0	0	115	62.5	
	ER + PR − HER2+	75	37.5	6	100	69	37.5	
Grade								0.426 ^1^
	1	12	6.5	0	0	12	6.5	
	2	70	37.5	1	16.7	69	37.5	
	3	40	21.2	1	16.7	39	21.2	
	Missing	68	34.5	4	66.7	64	34.8	
DCIS								0.666 ^2^
	Not present	128	66.8	5	83.3	123	66.8	
	Present	62	33.2	1	16.7	61	33.2	
Type NET								0.256 ^1^
	Tamoxifen (+/−GnRH)	25	13.6	0	0	25	13.6	
	Aromatase inhibitor	132	68.5	6	100	126	68.5	
	Missing	33	17.9	0	0	33	17.9	
Duration NET days								0.523 ^1^
	<180	98	51.1	3	50	95	51.6	
	181–270	65	34.2	3	50	62	33.7	
	>270	27	14.7	0	100	27	14.7	
Margin status								1.000 ^2^
	Negative	173	90.8	6	100	167	90.8	
	Positive	17	9.2	0	0	17	9.2	
Surgery type								0.407 ^2^
	Lumpectomy	74	39.7	1	16.7	73	39.7	
	Mastectomy	116	60.3	5	83.3	111	60.3	
Type of neo-adjuvant treatment								0.019 ^2^
	NET	150	80.4	2	33.3	148	80.4	
	NET+aHER2	40	19.5	4	66.7	36	19.6	

NST = No special type, ER = oestrogen receptor, PR = progesterone receptor, HER2 = human epidermal growth factor receptor 2, DCIS = ductal carcinoma in situ, NET = neoadjuvant endocrine therapy, NET+aHER2 = neoadjuvant endocrine + anti-HER2 therapy, GnRH = gonadotropin-releasing hormone, ^1^ = chi-squared test, ^2^ = Fisher’s exact test.

**Table 3 cancers-16-04188-t003:** Comparison of factors of patients with and without ypN0 status in patients with clinically node-negative disease.

		Total	%	ypN0	%	Non-ypN0	%	*p*-Value
Patients (N)		111	100	66	59.5	45	40.5	
Age years								0.007 ^1^
	<75	44	39.6	33	50.0	11	24.4	
	>75	67	60.4	33	50.0	34	75.6	
Menopausal status								0.647 ^2^
	Pre/peri	5	4.5	4	6.1	1	2.2	
	Post	106	95.5	62	93.9	44	97.8	
Histologic subtype								0.881 ^1^
	NST	91	82.0	55	83.3	36	80.0	
	Lobular	16	14.4	9	13.6	7	15.6	
	Other	4	3.6	2	3.0	2	4.4	
Clinical tumour status								0.602 ^1^
	cT1	33	29.7	22	33.3	11	24.5	
	cT2	64	57.7	36	54.5	28	62.2	
	cT3/4	14	9.6	8	12.1	6	13.3	
Multifocality								0.291 ^2^
	Unifocal	94	84.7	58	87.9	36	80.0	
	Multifocal	17	15.3	8	12.1	9	20.0	
Receptor status								0.278 ^1^
	ER + PR + HER2+	66	59.5	42	63.6	24	53.3	
	ER + PR − HER2+	45	40.5	24	36.4	21	46.7	
Grade								0.192 ^1^
	1	11	9.9	9	13.6	2	4.5	
	2	45	40.5	23	34.8	22	48.9	
	3	24	21.6	13	19.7	11	24.4	
	Missing	31	27.9	21	31.8	10	22.2	
DCIS								0.255 ^1^
	Not present	72	64.9	40	60.6	32	71.1	
	Present	39	35.1	26	39.4	13	28.9	
Type NET								0.829 ^1^
	Tamoxifen (+/−GnRH)	15	13.5	10	15.2	5	11.1	
	Aromatase inhibitor	79	71.2	46	69.7	33	73.3	
	Missing	17	15.3	10	15.2	7	15.6	
Duration NET days								0.174 ^1^
	<180	58	52.3	38	57.6	20	44.4	
	>180	53	47.6	28	42.4	25	55.6	
Margin status								0.520 ^2^
	Negative	100	90.1	58	87.9	42	93.3	
	Positive	11	9.9	8	12.1	36	6.7	
Surgery type								0.510 ^1^
	Lumpectomy	56	50.5	35	53.0	21	46.7	
	Mastectomy	55	49.5	31	47.0	24	53.3	
Type of neo-adjuvant treatment								0.055 ^1^
	NET	95	85.6	53	80.3	42	93.3	
	NET+aHER2	16	14.4	13	19.7	3	6.7	

NST = No special type, ER = oestrogen receptor, PR = progesterone receptor, HER2 = human epidermal growth factor receptor 2, DCIS = ductal carcinoma in situ, NET = neoadjuvant endocrine therapy, NET+aHER2 = neoadjuvant endocrine + anti-HER2 therapy, GnRH = gonadotropin-releasing hormone, ^1^ = chi-squared test, ^2^ = Fisher’s exact test.

**Table 4 cancers-16-04188-t004:** Comparison of factors of patients with and without ypN0 status in patients with clinically node-positive disease.

		Total	%	ypN0	*%*	Non-ypN0	%	*p*-Value
Patients (N)		79	100	9	11.3	70	88.7	
Age years								0.255 ^2^
	<75	22	27.8	4	44.4	18	25.7	
	>75	57	72.2	5	55.6	52	74.3	
Menopausal status								1.000 ^2^
	Pre/peri	3	3.8	0	0	3	4.3	
	Post	76	96.2	9	100	67	95.7	
Histologic subtype								0.659 ^1^
	NST	73	92.4	9	100	64	91.4	
	Lobular	5	6.3	0	0	5	7.1	
	Other	1	1.3	0	0	1	1.5	
Clinical tumour status								0.586 ^1^
	cT1	7	8.9	1	11.1	6	8.6	
	cT2	30	38.0	2	22.2	28	40.0	
	cT3/4	42	53.2	6	66.7	36	51.4	
Multifocality								1.000 ^2^
	Unifocal	54	68.4	6	66.7	48	68.6	
	Multifocal	25	31.6	3	33.3	22	31.4	
Receptor status								1.000 ^2^
	ER + PR + HER2+	49	62.0	6	66.7	43	61.4	
	ER + PR − HER2+	30	38.0	3	33.3	27	38.6	
Grade								0.095 ^1^
	1	1	1.3	0	0	1	1.4	
	2	25	31.6	4	44.4	21	30.0	
	3	16	20.3	4	44.4	12	17.1	
	Missing	37	46.8	1	11.1	36	51.5	
DCIS								0.715 ^2^
	Not present	56	70.9	6	66.7	50	71.4	
	Present	23	29.1	3	33.3	20	28.6	
Type NET								0.737 ^1^
	Tamoxifen (+/−GnRH)	10	12.7	1	11.1	9	12.9	
	Aromatase inhibitor	53	67.1	7	77.8	46	65.7	
	Missing	16	20.3	1	11.1	15	21.4	
Duration NET days								0.737 ^2^
	<180	39	49.4	5	55.6	34	48.6	
	>180	40	50.6	4	44.4	36	51.4	
Margin status								0.528 ^2^
	Negative	73	92.4	8	88.9	65	92.9	
	Positive	6	7.6	1	11.1	5	7.1	
Surgery type								0.198 ^2^
	Lumpectomy	18	22.8	4	44.4	14	20.0	
	Mastectomy	61	77.2	5	55.6	56	80.0	
Type of neo-adjuvant treatment								0.020 ^2^
	NET	55	69.6	3	33.3	52	74.3	
	NET+aHER2	24	30.4	6	66.7	18	25.7	

NST = No special type, ER = oestrogen receptor, PR = progesterone receptor, HER2 = human epidermal growth factor receptor 2, DCIS = ductal carcinoma in situ, NET = neoadjuvant endocrine therapy, NET+aHER2 = neoadjuvant endocrine + anti-HER2 therapy, GnRH = gonadotropin-releasing hormone, ^1^ = chi-squared test, ^2^ = Fisher’s exact test.

**Table 5 cancers-16-04188-t005:** ypT status with corresponding ypN status.

	ypN0	ypN1	ypN2	ypN3	ypNx	Total
**ypT0**	4	1	0	1	0	6
**ypT1**	41	17	4	2	5	69
**ypT2**	22	29	17	4	10	82
**ypT3**	5	3	3	1	1	13
**ypT4**	3	9	5	1	2	20
Total	75	59	29	9	18	190

**Table 6 cancers-16-04188-t006:** Univariable and multivariable logistic regression analyses for the occurrence of ypN0 status in patients with clinically node-negative disease.

		Univariabele			Multivariabele		
		Odds Ratio	95%CI	*p*-Value	Odds Ratio	95%CI	*p*-Value
Age years							
	<75	Ref.		0.008	Ref.		0.008
	>75	0.324	0.141–0.745		0.283	0.112–0.717	
Grade							
	1	Ref.		0.215	Ref.		0.093
	2	0.232	0.045–1.198		0.192	0.033–1.130	
	3	0.263	0.047–1.481		0.322	0.053–1.976	
	Missing	0.467	0.085–2.573		0.636	0.104–3.898	
Duration NET							
	<180	Ref.		0.175	Ref.		0.107
	>180	0.589	0.275–1.266		0.485	0.201–1.168	
Type of neo-adjuvant treatment							
	NET	Ref.		0.067	Ref.		0.016
	NET+aHER2 therapy	3.434	0.918–12.843		6.218	1.405–27.509	

NET = neoadjuvant endocrine therapy, NET+aHER2 = neoadjuvant endocrine + anti-HER2 therapy.

## Data Availability

The data that support the findings of this study are available from the authors, but restrictions apply to the availability of these data. Data are, however, available from the authors upon reasonable request and with permission from the Netherlands Cancer Registry.

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
