# Peer review of "Adding Anti-HER2 Therapy to Neoadjuvant Endocrine Therapy Seems Effective in Hormone Receptor and HER2 Positive Breast Cancer Patients Unfit for Chemotherapy: A Nationwide Population-Based Cohort Study"

_cancers, 2024, doi:10.3390/cancers16244188_

Round 1

Reviewer 1 Report

Comments and Suggestions for Authors

This manuscript evaluates the efficacy of NET combined with anti-HER2 therapy versus NET alone in HR+/HER2+ breast cancer patients who are unfit for chemotherapy using a national retrospective cohort. The results indeed showed that the combination of NET and aHER2 resulted in significantly higher rates of ypT0 (10% vs. 1.3%) and ypN0, especially in clinically node-positive patients, therefore indicating a potential clinical benefit. This is relevant, especially for elderly or comorbid patients, since the current study focuses on a less toxic regimen. The use of a national cancer registry further strengthens the generalizability of this study.

Limitations that should be addressed before publication include the following:

Latest Evidence of SLNB after neoadjuvant chemotherapy: The authors should discuss more about the latest evidence after sentinel lymph node biopsy in cN+ breast cancer patients after neoadjuvant chemotherapy. In fact, patients treated with less invasive axillary surgery apparently present better long-term oncological outcomes. Combination of NET and aHER2 could potentially lead to more SLNBs. Please cite PMID: 39335140 to enhance your Discussion section and improve the quality of your manuscript, overall.

Small Sample Size: The sample size is very small, with only 190 patients included and only 40 receiving NET+aHER2, which limits statistical power. Did the authors do a power analysis? If yes, please include it in the Methods section. If no, please explain why.

Retrospective Design: Inherent biases and confounding by indication, such as missing performance status and cardiac function data, make causal inferences less reliable.

No Survival Data: The absence of analysis of breast cancer-specific or overall survival reduces the clinical effect of the results.

Lack of Dual-HER2 Blockade: It also does not present the efficacy of contemporary dual-HER2 blockade regimens.

No Quality of Life Assessment: Further, toxicity and QoL data, which are very important for elderly patients, are lacking.

Guideline Variability: Treatment choices might be influenced by changes in guidelines through the period under study.

Reviewer 2 Report

Comments and Suggestions for Authors

The manuscript provides valuable insights into the potential role of combining anti-HER2 therapy with neoadjuvant endocrine therapy for HR+/HER2+ breast cancer patients unfit for chemotherapy. Despite some limitations, the study addresses an unmet need and offers clinically significant findings. Addressing the weaknesses outlined below would strengthen the manuscript and enhance its impact on clinical practice and future research.

General comments:

Strengths:

The manuscript addresses an important clinical gap in the treatment of HR+/HER2+ breast cancer patients who are unfit for chemotherapy. These patients represent a challenging population, and the study provides much-needed data on alternative neoadjuvant systemic treatment (NST) options. The use of data from the Netherlands Cancer Registry ensures that the findings are based on a large, real-world population, enhancing the generalizability of the study. The analysis of ypT0 and ypN0 rates as primary endpoints is appropriate, as these serve as established surrogate markers for treatment efficacy in the neoadjuvant setting. The results demonstrate that the addition of anti-HER2 therapy (NET+aHER2) significantly improves ypT0 and ypN0 rates compared to NET alone, particularly in both cN+ and cN- patients. These findings are clinically relevant and provide a potential therapeutic alternative for this patient population.

Weaknesses and Areas for Improvement:

  1. Small Sample Size in NET+aHER2 Group:

The sample size of patients treated with NET+aHER2 (n=40) is relatively small compared to the NET group (n=150), limiting the statistical power of the analysis. This could affect the robustness of the conclusions drawn and should be acknowledged more prominently in the limitations section. Expand the discussion of limitations to address the small sample size in the NET+aHER2 group and the potential impact of unmeasured confounders.

  1. Lack of Detailed Treatment Information:

The manuscript does not provide sufficient details on the specific types and regimens of anti-HER2 therapy used in the NET+aHER2 group. The variability in treatment protocols may have impacted the outcomes and should be clarified. Include detailed information on the specific anti-HER2 therapies used (e.g., trastuzumab, pertuzumab) and their dosages to provide better context for the treatment regimens in the NET+aHER2 group.

  1. Confounding Variables:

While multivariable logistic regression was used, the manuscript lacks a comprehensive discussion of potential unmeasured confounders (e.g., comorbidities, performance status, and other patient characteristics) that may have influenced treatment selection and outcomes.

  1. Focus on Surrogate Endpoints:

While ypT0 and ypN0 rates are important surrogate endpoints, the manuscript would benefit from additional data on long-term clinical outcomes, such as overall survival (OS) and disease-free survival (DFS), to provide a more complete picture of treatment efficacy.

  1. Limited Exploration of Molecular Subtypes:

HR+/HER2+ breast cancer is a heterogeneous disease. The study does not address whether intrinsic subtypes or specific biomarkers influence response rates, which could add depth to the findings. Consider performing exploratory subgroup analyses based on intrinsic subtypes or specific biomarkers, if feasible.

  1. Age-Restricted Population:

The median age of 77 suggests that the findings are most applicable to older patients. This should be explicitly noted in the discussion, as younger patients unfit for chemotherapy may have different clinical characteristics and responses. Provide a more detailed exploration of age-related findings and how these results might apply to younger or more diverse patient populations.

  1. Highlight the need for prospective studies to validate these findings and evaluate long-term outcomes, such as OS and DFS. Add a section discussing the clinical implications of the findings, including how they might influence treatment guidelines for HR+/HER2+ patients unfit for chemotherapy.

Specific Comments:

1.        Line82-84, revise this sentence “In all these neodjuvant trials patients were treated with anti-HER2 therapy in combination with ET. Hence it is unknown whether the use of ET combined with anti-HER2 therapy (ET+aHER2) is effective in the neoadjuvant setting.”

2.        Line 185, “NET and four of whom were treated”, “four” should be “19”.

3.        Line 191-192, “patients treated with NET (p=0.047) (see table 3a).” In Table 3a, do not see this “p=0.047”.

4.        Line 220-221, “In cN+ patients (N=79), univariable analyses showed that a ypN0 status tended to be associated with type of neoadjuvant treatment (p = 0.055).”  This p = 0.055 was for the cN- patients, not cN+ patients, as shown in Table 3A.

Round 2

Reviewer 1 Report

Comments and Suggestions for Authors

The manuscript can be accepted in the present form